# RelChaNet: Neural Network Feature Selection using Relative Change Scores

## Abstract

There is an ongoing effort to develop feature selection algorithms to improve interpretability, reduce computational resources, and minimize overfitting in predictive models. Neural networks stand out as architectures on which to build feature selection methods, and recently, neuron pruning and regrowth have emerged from the sparse neural network literature as promising new tools. We introduce RelChaNet, a novel and lightweight supervised feature selection algorithm that uses neuron pruning and regrowth in the input layer of a dense neural network. For neuron pruning, a gradient sum metric measures the relative change induced in a network after a feature enters, while neurons are randomly regrown. We also propose an extension that adapts the size of the input layer at runtime. Extensive experiments on nine different datasets show that our approach generally outperforms the current state-of-the-art methods, and in particular improves the average accuracy by 2% on the MNIST dataset. Our code is available in the supplementary material.

## 1 Introduction

Feature selection is an elemental task in predictive modelling. It can serve to reduce computational resources, improve interpretability by highlighting important features, or improve predictive performance by reducing overfitting (Li et al., 2018). To further these goals has been the driving motivation of large recent efforts to improve existing and develop new feature selection algorithms. Feature selection algorithms can be categorized into embedded, wrapper, and filter approaches. Embedded methods select features during training of a predictive model, such as linear regression (Tibshirani, 1996) or neural networks (Lemhadri et al., 2021). Wrapper approaches also work around a specific predictive model, but treat it as a black box with the feature set as a hyperparameter, e.g., via particle swarm optimization (Rostami et al., 2021). Filter approaches select feature sets without being tailored around a predictive model, but using information-theoretic measures. They include, for example, statistical tests of the relationship between the feature and the outcome (Bommert et al., 2020).

Neural networks have a great ability to capture nonlinear relationships and offer many entry points for slightly modifying their architecture or training algorithm to build successful embedded feature selection methods. To decide on the utility of an input neuron, approaches added gates in the input layer (Yamada et al., 2020), added residual connections to the output (Lemhadri et al., 2021), or added gradients with respect to data changes to the loss (Cherepanova et al., 2023).

Feature selection in neural networks translates to aiming for a sparse input layer and is therefore a special case of sparse neural networks (Hoefler et al., 2021). Recently, it was shown that sparse neural network training (Mocanu et al., 2018; Evci et al., 2020) can be adapted to achieve a dominant feature selection performance (Liu et al., 2024; Atashgahi et al., 2024; Sokar et al., 2024). However, we have identified potential improvements to enhance the network's ability to detect important features and make it easier for regrown neurons to compete with established neurons during training.

In this paper, we introduce RelChaNet, a novel neural **net**work feature selection algorithm using **rel**ative **cha**nge scores. It applies neuron pruning and regrowth in the input layer of a dense neural network based on a relative change metric shown in Figure 1. Our main contributions are:

1. The RelChaNet feature selection algorithm, which has two key hyperparameters that allow it to adapt to the characteristics of the dataset used. It addresses two identified drawbacks by giving candidates multiple mini-batches of time to show their potential relevance in the

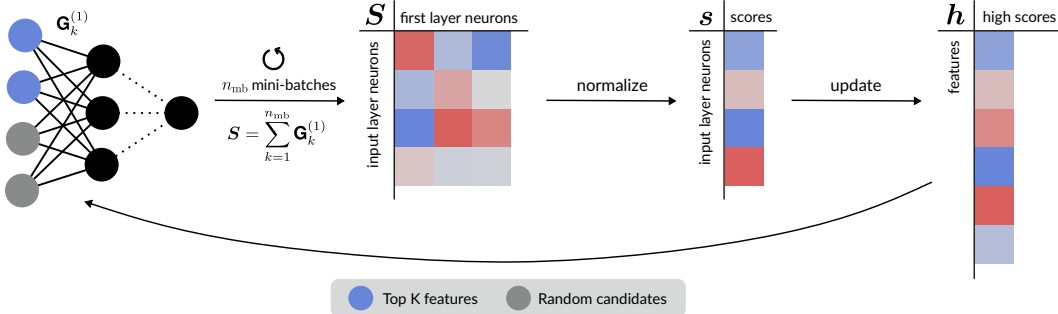

Figure 1: Illustration of the relative change score calculation embedded in RelChaNet, Algorithm 1. We consider a neural network with an input layer size equal to the number of features to select, $K$, plus additional candidate features. Over several mini-batches, determined by the hyperparameter $n_{\mathrm{mb}}$, the first layer gradients $\mathbf{G}_k^{(1)}$ are accumulated in a matrix $\boldsymbol{S}$. Next, these gradient sums are normalized by taking the $L^1$ norm with respect to each input neuron, followed by z-standardizing the resulting vector to produce a score vector $\boldsymbol{s}$. The scores of all candidates are then used to update the high scores $\boldsymbol{h}$. Finally, features among the top $K$ high scores remain in the network, while the other features are randomly redrawn. Before continuing training, the first layer weights of candidate features are reinitialized.

> network, and by comparing relevance as determined by the change induced rather than by absolute weights.

2. A version of the algorithm that can adapt the input layer size during runtime, making the algorithm less sensitive to one of its hyperparameters.

3. An evaluation of the approach on nine diverse datasets, demonstrating that it generally outperforms the current state-of-the-art.

The structure of this paper is as follows: We begin with a review of related work, particularly focusing on neural network-based methods. Next, we present the RelChaNet algorithm and its extension with an adaptive input layer size. We then conduct an extensive experiment to empirically evaluate our approach. Finally, we perform auxiliary analyses to investigate its design parameters and computational efficiency.

## 2 BACKGROUND AND RELATED WORK

In this section, we introduce the feature selection problem within the framework of neural networks and review previous solution approaches. Most approaches slightly modify a dense neural network architecture or the loss function. Recently, successful approaches have been taken from the framework of sparse neural networks.

**Feature selection in neural networks.** We consider the task of selecting a set of $K$ features that are most valuable for making accurate predictions in a supervised learning setting. Specifically for neural networks, we can express this task using $L^0$ regularization of the first layer network weights. Accordingly, we want to optimize the network under the condition that only $K$ input neurons are active, i.e., have any non-zero adjacent weights. If we consider a neural network with one input neuron for each feature $i \in \{1, \ldots, N\}, N > K$, we can express the feature selection task as finding a specific set of network weights $\mathbf{W}$ that fulfills

$$\operatorname*{arg\,min}_{\mathbf{W}} \left\{ \mathcal{L}(\mathbf{W}) \mid \#\{i \mid ||\mathbf{W}_{i.}^{(1)}||_1 > 0\} = K \right\} \tag{1}$$

where $\mathcal{L}(\mathbf{W})$ represents evaluating the loss function $\mathcal{L}$ using the data and network weights, and $\mathbf{W}_{i.}^{(1)}$ is the vector of outgoing first layer weights from input neuron $i$. The key challenge in solving this task is to implement an effective $L^0$ regularization. Exact solutions are computationally prohibitive and become intractable in high-dimensional settings (Yamada et al., 2020). Consequently, the related work discussed below uses various approximations to address this challenge.

**Dense neural networks.** There are several methods embedded in dense neural networks for feature selection. A common property is that the number of active neurons is not strictly enforced before model convergence. Instead, selection is gradual, starting with a full input layer of $N$ neurons and reducing active neurons during training. This approach makes it easier to identify complex interactions between features, at the cost of increased computational complexity. Stochastic gates (Yamada et al., 2020) approach the $L^0$ regularization by adding a gate to each input layer neuron. For each gate, a trainable parameter controls the probability of a feature being active. The LassoNet (Lemhadri et al., 2021) adds a residual connection from each input layer neuron to the network output. The absolute sizes of these $N$ residual weights are added to the loss function and for each feature $i$ individually represent a bound on the size of the corresponding first layer weights, $||\mathbf{W}^{(1)}_{i\cdot}||_1$. A less invasive approach is DeepLasso (Cherepanova et al., 2023), which adds the gradient with respect to changes in the input data to the loss function. This encourages the network not to use some features during training, rendering the corresponding input neuron inactive.

**Sparse neural networks.** Sparse neural networks keep a large fraction of the weights throughout the network at 0 to reduce memory requirements or training time (Hoefler et al., 2021). One method to achieve this is structured sparsity, such as neuron pruning, where all of a neuron's outgoing weights are set to 0. Metrics for deciding which neurons to prune include the magnitude of the outgoing weights or the sensitivity of the output to the neuron. Neurons can also be periodically regrown, based on criteria such as the size of gradients or adjacent weights. Molchanov et al. (2019) propose a neuron/filter pruning method that calculates a score across mini-batches, similar to our approach. However, their method is not specific to the input layer, calculates the product of weight and gradient, and does not involve regrowing neurons or reusing a score later in training. GradEnFS (Liu et al., 2024) uses sparse neural networks for feature selection. Similar to DeepLasso, it measures the importance of neurons based on how sensitive the loss is to changes in the input neurons. After the model converges, it selects the top $K$ features based on neuron importance. We see this selection procedure as a disadvantage because no specific sets of $K$ features are assessed during training.

Since pruning the input layer reduces the number of active neurons, as required in Equation 1, methods that do so are promising for feature selection. NeuroFS (Atashgahi et al., 2023) extends adaptive sparse neural network training, which utilizes weight pruning (Mocanu et al., 2018; Evci et al., 2020), by incorporating input layer neuron pruning. Input neurons are pruned after each epoch based on the magnitude of their outgoing connections, $||\mathbf{W}^{(1)}_{i\cdot}||_1$. To regrow an input neuron, NeuroFS calculates the absolute gradients of all currently pruned first layer weights. Neurons are then regrown based on the largest absolute gradient among their adjacent weights. During training, the number of active neurons in the input layer is continuously reduced. After training, the input neurons with the largest outgoing connections among the remaining active neurons are selected.

We generally observe two drawbacks in gradient-based regrowing and absolute weight-based pruning for feature selection. Firstly, in the regrowing procedure, features need to signal their importance through high adjacent gradients before the network makes any adjustments for them. However, the network might take longer, e.g., multiple mini-batches, to recognize the importance of a feature, especially if it is involved in complex interactions with other features. Secondly, in later training epochs, the absolute weights of regrown neurons are compared to those of longer established neurons. In consequence, features are compared while being given different times to grow their weights. To mitigate both of these drawbacks, we propose to regrow features randomly and to use a metric of the change a feature induces in the network over the first few mini-batches after it enters the network for pruning.

## 3 THE RELCHANET ALGORITHM

We propose the RelChaNet algorithm for supervised feature selection using neural networks. RelChaNet computes a score for each input neuron by aggregating gradients over mini-batches. These scores are normalized and used to update a high score vector, which guides feature selection (see Figure 1 for an illustration). This section walks through the pseudocode in Algorithm 1 and explains its rationale. RelChaNet is implemented using PyTorch (Paszke et al., 2019) and is available as a Python package in the supplementary material.

---

**Algorithm 1** RelChaNet

---

1: **Input.** Dataset with $N$ features, number of selected features $K$, number of first hidden layer neurons $n_{\text{hidden}}$. Hyperparameters: Ratio of candidate features $c_{\text{ratio}}$, number of mini-batches $n_{\text{mb}}$

2: **Initialize.** Number of candidate features $K_c = \text{round}(c_{\text{ratio}}(N - K))$. Network with input layer size $K + K_c$. Randomly choose features to populate the input layer, $I_{\text{input}} = I_{\text{cands}} = \text{Rand}(\{1, \ldots, N\}, K + K_c)$. Score vector $\boldsymbol{s} \in \mathbb{R}^{K+K_c}$, high score vector $\boldsymbol{h} \in \mathbb{R}^N$. First layer gradients $\mathbf{G}^{(1)}$ and gradient sum matrix $\boldsymbol{S}$: $\mathbf{G}^{(1)}, \boldsymbol{S} \in \mathbb{R}^{(K+K_c)\times n_{\text{hidden}}}$

3: **while** training not stopped **do**

4:     $\boldsymbol{S} = 0$

5:     **for** $n_{\text{mb}}$ mini-batches **do**

6:        *Feed-forward step and backpropagation using a mini-batch of data*

7:        $\boldsymbol{S} = \boldsymbol{S} + \mathbf{G}^{(1)}$

8:     **end for**

9:     $\boldsymbol{s}_i = \sum_{j=1}^{n_{\text{hidden}}} |\boldsymbol{S}_{ij}|$ for $i \in \{1, \ldots, K + K_c\}$

10:    Normalize $\boldsymbol{s} = (\boldsymbol{s} - \text{Mean}(\boldsymbol{s}))/\text{SD}(\boldsymbol{s})$

11:    Update high scores $\boldsymbol{h}_{I_{\text{cands}}} = \max(\boldsymbol{h}_{I_{\text{cands}}}, \boldsymbol{s}_{\text{cands}})$, where cands is the set of input neurons corresponding to $I_{\text{cands}}$

12:    Identify top features $I_{\text{top}} = \{i \in \{1, \ldots, N\} \mid \boldsymbol{h}_i \geq \text{quantile}(\boldsymbol{h}, 1 - K/N)\}$

13:    Draw new candidates $I_{\text{cands}} = \text{Rand}(\{1, \ldots, N\} \setminus I_{\text{top}}, K_c)$

14:    Update features that populate the input layer $I_{\text{input}} = I_{\text{top}} \cup I_{\text{cands}}$

15:    Initialize first layer weights $\mathbf{W}^{(1)}_{\text{cands.}} = U(-10^{-8}, 10^{-8})$. Initialize the optimizer

16: **end while**

---

**Architecture and initialization.** The algorithm uses a multi-layer perceptron (MLP) with a feed-forward architecture and is integrated into the backpropagation training using the Adam optimizer (Goodfellow et al., 2016; Kingma & Ba, 2015). This implies the adoption of the hyperparameters of learning rate, batch size, and number of hidden layers and their sizes. The size of the input layer is based on the desired number of selected features $K$ plus a percentage $c_{\text{ratio}}$ of the remaining features, $K_c$, which will be referred to as candidates.

**Relative change scores.** Steps 5-10 calculate the relative change scores $\boldsymbol{s}$, where gradient sums for each input neuron are aggregated and normalized to reflect their relative contribution across the last $n_{\text{mb}}$ mini-batches (see also Figure 1). Instead of gradient sums, one could also use weight changes as a relative change metric in Steps 5-8, which we compare in an ablation study in Section 4.2.

**Input layer rotation.** Steps 11–15 dynamically update the input layer by selecting a combination of top features and new candidates. The relative change scores computed earlier are used to identify the top $K$ features (Step 12), ensuring they remain in the input layer. Additional candidate features are randomly sampled from the remaining features (Step 13). Together, these form the input layer (Step 14). To avoid symmetry issues during training, the weights of candidate features are reinitialized to small random values (Step 15), following best practices in neural network initialization (Goodfellow et al., 2016). This rotation ensures that feature selection is iteratively refined based on relevance.

**Key mechanism.** Our algorithm approaches the $L^0$ regularization task laid out in Equation 1 by stabilizing the high score vector $\boldsymbol{h}$. At the time of each input layer rotation, the network is forced to adhere to the criterion of only $K$ active features, after which it gets to assess additional candidates again for a few mini-batches. The high scores $\boldsymbol{h}$, since they preserve information over time, allow a comparison of the entry performance of candidates with the entry performance of features that entered epochs ago. Specifically, in later epochs of training, good candidates do not need to surpass the absolute first layer weights of the more established neurons.

**Random Regrowth.** Random regrowth offers a key advantage over metric-based methods, such as gradient-based selection, by giving candidates multiple mini-batches to demonstrate their relevance. It facilitates the inclusion of features that do not have a straightforward relationship with the output but contribute to complex patterns that only emerge over time. Additionally, by combining random

---

**Algorithm 2** RelChaNet flex

---

1: **Initialize:** Loss before the last change of the network size $l_{\text{change}}$, running loss $l$. Set the input layer size change direction to shrink.
2: **if** $l$ has not decreased for 10 rotations **then**
3:    **if** $l > l_{\text{change}}$ **then**
4:       Change the direction: shrink $\leftrightarrow$ grow
5:    **end if**
6:    $l_{\text{change}} = l$
7:    **if** direction is shrink **then**
8:       $c_{\text{ratio}} = \max(\frac{1}{2} c_{\text{ratio}}, \frac{1}{5} \frac{K}{N-K})$
9:    **else if** direction is grow **then**
10:      $c_{\text{ratio}} = \min(2 \, c_{\text{ratio}}, 1)$
11:    **end if**
12: **end if**

---

regrowth with a reduced input layer size, we can decrease computation compared to methods that retain all features in the input layer for regrowth selection. One drawback of random regrowth is that it may take several rotations before all features have the opportunity to enter the input layer at least once or before interacting feature sets are included together in the input layer. Increasing the $c_{\text{ratio}}$ hyperparameter can mitigate this issue by enlarging the input layer, allowing more features to be included at a time. However, this approach introduces more noise into the network, as a larger portion of the network is frequently reset to zero, potentially disrupting the learning process of relevant features. Consequently, the choice of $c_{\text{ratio}}$ reflects a tradeoff between exploration and exploitation. A potential solution to this challenge is to adapt $c_{\text{ratio}}$ dynamically during training, which we explore below.

## 3.1 ADAPTIVE NETWORK SIZES

To address sensitivity to the $c_{\text{ratio}}$ hyperparameter, we introduce RelChaNet flex, which dynamically adjusts the input layer size during training based on the behavior of the loss function. It extends Algorithm 1 between Steps 12 and 13, i.e., prior to selecting new candidate features, and is detailed in pseudocode in Algorithm 2, RelChaNet flex.

**Key mechanism.** RelChaNet flex monitors the running loss, $l$, and compares it with the loss recorded at the time of the last input layer size change, $l_{\text{change}}$. If the loss stagnates (i.e., does not decrease for a fixed number of rotations), the algorithm adjusts the input layer size. Specifically:

1. Direction adjustment: If the loss increases compared to $l_{\text{change}}$, the direction of change (shrink or grow) is reversed
2. Size adjustment: Depending on the direction, $c_{\text{ratio}}$ is halved or doubled, bounded by predefined limits. The upper limit of $c_{\text{ratio}} = 1$ represents using the maximum number of candidates, $N - K$, while the lower limit ensures a minimum input layer size of $\frac{6}{5} K$.

**Rationale.** A well-balanced input layer size allows the network to explore a sufficient pool of candidate features in the presence of random regrowth. Shrinking the input layer promotes stability, while growing it enables exploration of additional candidates. The dynamic adjustment ensures that the network can escape suboptimal configurations.

**Practical considerations.** The running loss $l$ as well as the loss at the time of input layer change, $l_{\text{change}}$, can be either a training or validation loss, depending on whether the algorithm is used with a validation set. In our experiments, we use a validation set, which is detailed in Appendix A.2.

## 4 EXPERIMENTS

In this section, we conduct an empirical evaluation of our proposed algorithms structured into a main experiment and additional analyses. To conserve computational resources, we replicate the

Table 1: Dataset dimensions and domain

|  | Cases | Features | Domain | Reference |
|---|---|---|---|---|
| **Long Datasets** |  |  |  |  |
| COIL-20 | 1440 | 1024 | Image | Nene et al. (1996) |
| HAR | 10299 | 561 | Smartphone Sensor | Anguita et al. (2013) |
| ISOLET | 7797 | 617 | Speech | Fanty & Cole (1990) |
| MNIST | 70000 | 784 | Image | Deng (2012) |
| Fashion-MNIST | 70000 | 784 | Image | Xiao et al. (2017) |
| USPS | 9298 | 256 | Image | Hull (1994) |
| **Wide Datasets** |  |  |  |  |
| ARCENE | 200 | 10000 | Genomics | Guyon et al. (2004) |
| GLA-BRA-180 | 180 | 49151 | Genomics | Sun et al. (2006) |
| Prostate-GE | 102 | 5966 | Genomics | Nie et al. (2010) |

experimental setup of Atashgahi et al. (2023)[1] and compare our results with those of nine state-of-the-art baseline methods reported in their work. The compared baseline methods and the RelChaNet implementations are described in Appendix A. Code for replicating the main experiment is available in the supplementary material.

The datasets used and their dimensions are shown in Table 1. We categorize datasets as long if they have more cases than features, and vice versa as wide. The datasets all represent classification tasks and span different content domains, including speech processing (ISOLET), image recognition (MNIST), and smartphone sensor data (HAR). They are all freely available. To provide a more comprehensive evaluation, we include an auxiliary experiment on four additional datasets, detailed in Appendix C. These include two additional long datasets (CIFAR-10 and CIFAR-100) to explore performance on complex prediction tasks and two additional wide datasets (BASEHOCK and SMK).

To ensure a fair comparison between embedded and filter methods, all experimental conditions include downstream learners. Initially, the data is split into training and test sets. Feature selection is performed using the training data, followed by training a downstream predictive model on the training data using only the selected features. The accuracy of the downstream learner is then evaluated on the test data. The number of selected features, $K$, varies among 25, 50, 75, and 100[2]. The downstream learners are classifiers based on a Support Vector Machine (SVM, Chang & Lin, 2011), K-Nearest Neighbors (KNN), and ExtraTrees (ET, Geurts et al., 2006). The SVM classifier is used for all values of $K$, while KNN and ET are only used for $K = 50$. Each condition is run five times. Experiments are conducted on an NVIDIA GeForce RTX 3060 GPU with 6GB of memory.

## 4.1 RESULTS

Figure 2 presents a comparison of the accuracies achieved using our methods ("RCN" and "RCN flex") against the top baseline methods for the SVM downstream learner. The average accuracy by dataset is shown for all methods in Figure 3. Detailed results for each dataset, method, and value of $K$ are provided in Table 2 in Appendix B.

According to the results, our approaches significantly outperform the baseline methods for most long datasets (first six panels in the plots). In particular, for the MNIST dataset, our flex approach achieves an average accuracy of 96.3%, significantly improving on the best previous result of 94.3%. Our methods demonstrate comparable performance to the baseline methods for the wide datasets (last three panels in the plots), but fall slightly behind for the GLA-BRA-180 dataset. The RCN and RCN

---

[1]This includes code for data preprocessing, train-test split, and downstream learners, which is available at https://github.com/zahraatashgahi/NeuroFS. The performance of the downstream learners using all features was compared with the reported values to ensure accurate replication of the experiment setup. As detailed in our supplementary material, this was unsuccessful for the BASEHOCK and SMK datasets, which are therefore omitted from the experiment.

[2]Atashgahi et al. (2023) also used higher values for $K$ which are omitted in this study since there was little variance in the results between the different methods.

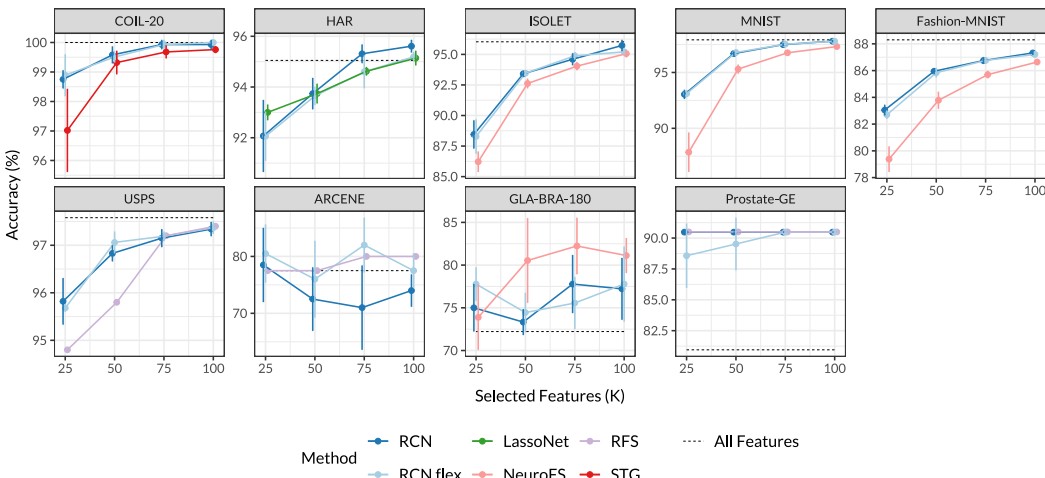

Figure 2: Resulting accuracy for the studied methods by dataset and number of selected features $K$ using the SVM downstream learner. Our proposed methods are "RCN" and "RCN flex". For visual clarity, only the baseline method with the highest average accuracy for each dataset is shown. "All Features" is the accuracy using all features in the dataset. Error bars indicate the standard deviation. Results for the baseline methods are reproduced from Atashgahi et al. (2023).

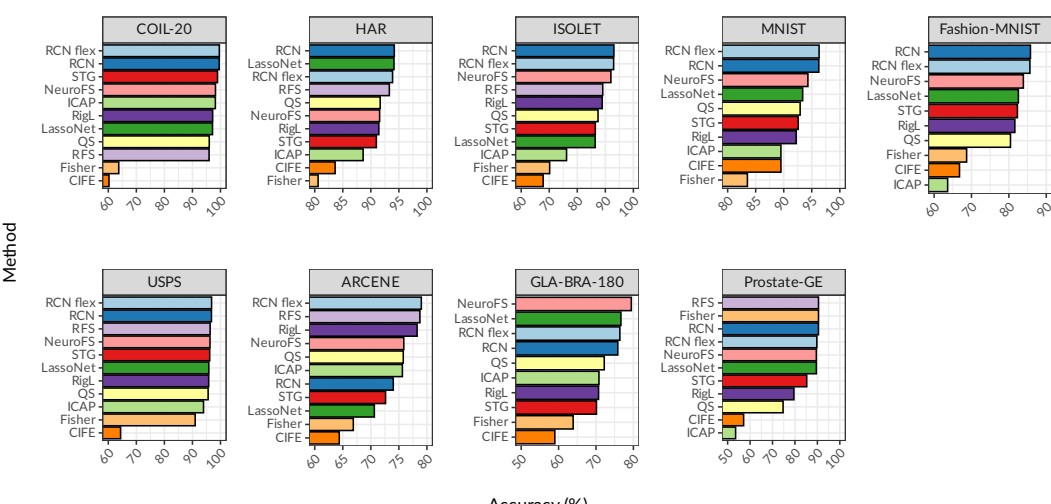

Figure 3: Average accuracy by dataset for the studied methods using the SVM downstream learner. Our proposed methods are "RCN" and "RCN flex". Results for the baseline methods are reproduced from Atashgahi et al. (2023).

flex approaches perform similarly in all conditions except for the ARCENE dataset, where the RCN approach performs notably worse than the top baselines.

We also evaluated two additional downstream learners, KNN and ET, under the condition of $K = 50$ selected variables (see Table 3 in Appendix B). The results are very similar to those obtained with the SVM classifier, indicating that the selected feature sets are valuable across multiple downstream learners.

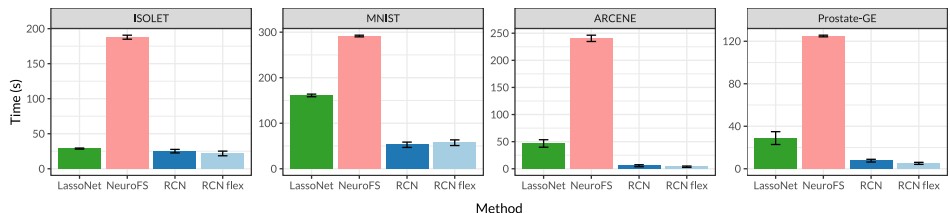

Figure 4: Wall-clock run time for the studied methods by dataset. All conditions use $K = 50$ selected features and are repeated five times. The error bars indicate the standard deviation.

## 4.2 ADDITIONAL ANALYSES

In this section, we highlight some additional aspects to give a more complete picture of RelChaNet. We include a comparison of the computational efficiency with similar methods, an ablation study of the impact of the chosen change metric, and an investigation of the impact and feasible ranges of hyperparameters.

**Computational efficiency.** We examine the comparative computational costs with two other approaches, NeuroFS and LassoNet. Both are well-performing sparse and dense neural network based methods, respectively. One drawback of our approach is that, since candidate features are chosen randomly, it generally requires more training epochs than other approaches to ensure that all features get the chance to enter the network. This motivates comparing the overall runtime of the approaches.

We measure the wall-clock time for selecting $K = 50$ features, using two wide and two long datasets, with settings otherwise as in the main experiment. For NeuroFS, we use the setup from the original publication: a 3-layer sparse MLP with 1000 neurons in each layer, limiting the training epochs to 100. For LassoNet, we use the same MLP architecture as for RelChaNet, i.e., one hidden layer with 100 neurons. We keep all other settings at the LassoNet package defaults[3]. Each condition is run five times.

The results are shown in Figure 4. The RCN and RCN flex approaches have comparable runtimes, both demonstrating significantly greater efficiency than NeuroFS across the studied datasets. Additionally, RCN is more efficient than LassoNet in three out of four conditions. One explanation for RelChaNet's efficiency is that its higher number of required epochs is offset by a relatively small computational overhead. However, NeuroFS utilizes binary masks to implement sparse networks, and future advancements in hardware optimized for sparse matrix computations could improve its efficiency.

**Ablation study: Change metrics.** We compare the performance of RelChaNet under different change metrics. Specifically, we evaluate the gradient sums used in RelChaNet against using weight changes or absolute weights. In both cases, the calculation of $S$ is modified immediately before Step 9 of Algorithm 1. For the weight changes, we set $S = \mathbf{W}^{(1)} - \mathbf{W}^{(1)}_{\text{old}}$, where $\mathbf{W}^{(1)}_{\text{old}}$ are the first layer weights at the time of the last rotation. For the absolute weights, we simply set $S$ equal to the first layer weights, $S = \mathbf{W}^{(1)}$. We use four datasets, two long and two wide, and $K = 50$ selected features, keeping all other properties the same as in the main experiment.

Figure 5 shows the results. For the long datasets (left two panels), the gradient sums and weight changes perform similarly, surpassing the performance of absolute weights. For the wide datasets (right two panels), the gradient sums show superior performance, while the other two approaches exhibit similar effectiveness. In summary, under the studied conditions, gradient sums are the most effective metric for measuring relative change within the RelChaNet algorithm.

**Impact of hyperparameters.** We investigate the role of the hyperparameters $c_{\text{ratio}}$ and $n_{\text{mb}}$. Generally, $c_{\text{ratio}}$ determines the percentage of features included in the network in addition to the $K$ selected features, while $n_{\text{mb}}$ specifies the number of mini-batches after which scores are computed

---

[3]The LassoNet package is available at https://github.com/lasso-net/lassonet.

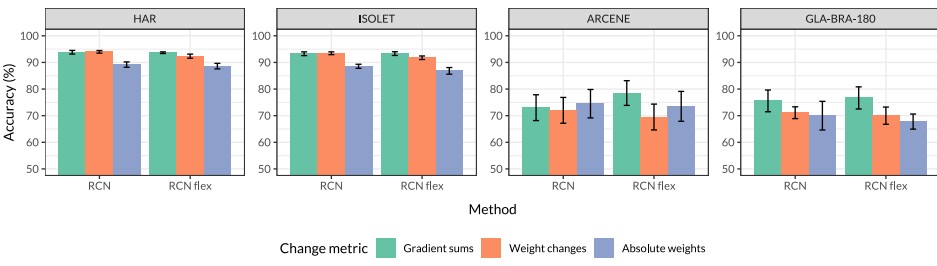

Figure 5: Resulting accuracy for the studied change metrics by RCN method and dataset for $K = 50$ selected features using the SVM downstream learner. The error bars indicate the standard deviation.

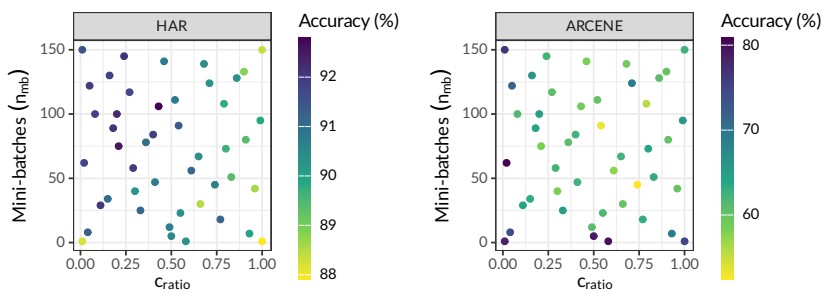

Figure 6: Accuracy using RelChaNet feature selection by hyperparameters $c_{\text{ratio}}$ and $n_{\text{mb}}$ for two datasets and $K = 25$ selected features. Each point represents the average of three runs.

and features are rotated. We use a long and a wide dataset, HAR and ARCENE, $K = 25$, and keep all other properties consistent with the main experiment. We let $c_{\text{ratio}}$ vary between 0.01 and 1 and $n_{\text{mb}}$ between 1 and 150. As studied hyperparameter sets we include the two configurations from our experiment: ($c_{\text{ratio}} = 0.2$, $n_{\text{mb}} = 100$) for the long datasets and ($c_{\text{ratio}} = 0.5$, $n_{\text{mb}} = 5$) for the wide datasets. Additionally, we include the four corners of the hyperparameter space and draw 40 pseudo-random sets of configurations from a Halton sequence. Each resulting condition is run three times, and the accuracy is averaged.

The results are illustrated in Figure 6. For the long HAR dataset (left panel), the combination of low $c_{\text{ratio}}$ and high $n_{\text{mb}}$ yields strong results. In contrast, for the ARCENE dataset (right panel), configurations with low $n_{\text{mb}}$ generally perform well. A combination of low $c_{\text{ratio}}$ and higher $n_{\text{mb}}$ may also be effective. This highlights that hyperparameters must be selected differently for different datasets, with a comparatively narrower range working well for wide datasets.

## 5 DISCUSSION

In this paper, we introduce a novel feature selection algorithm aimed at enhancing the predictive performance and interpretability of predictive models. Our approach incorporates neuron pruning and regrowth from the sparse neural network literature into a dense neural network framework. RelChaNet uses a relative change metric for pruning, which measures the relative change induced in a network after a feature enters, while neurons are randomly regrown. Extensive experiments demonstrate that our method, along with an extension featuring an adaptive input layer, consistently outperforms state-of-the-art techniques on datasets with more cases than features. For datasets with more features than cases, its performance is comparable to previous approaches. While the adaptive version has theoretical advantages and performs better on one dataset, the base algorithm stands out for its simplicity and competitive performance in most scenarios.

The primary limitation of our approach lies in its theoretical disadvantage in computational efficiency. This is due in part to the reliance on a dense network, which typically has higher computational

training costs than sparse networks with the same number of layers and neurons. Additionally, regrowing neurons randomly necessitates either a large input layer or longer training. However, our experiment demonstrates that these challenges can be mitigated by employing a small neural network architecture without compromising feature selection performance. Furthermore, the efficiency was found to be competitive with another dense approach. It is important to note, however, that this may not generalize to scenarios beyond those studied.

We see many potential directions for future research. One avenue is to integrate our pruning and regrowth protocol into sparse neural networks. This could be applied to the input layer for feature selection, or extended to other layers for general sparse neural network training. Another direction is to explore the utility of our approach for interpretable machine learning. For instance, the values in the high score vector $h$ could be evaluated as a measure of variable importance.

## REPRODUCIBILITY STATEMENT

We include the source code of our method in the form of a Python package, as well as code to reproduce the main experiment results, in the supplementary material.

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

## A EXPERIMENTAL SETUP

### A.1 BASELINES

The methods compared against our approach are as follows. Their specific implementations are detailed in Atashgahi et al. (2023):

- Fisher Score (Gu et al., 2011): A classic filter method that selects feature sets based on their ability to separate data points.
- CIFE (Conditional Infomax Feature Extraction, Lin & Tang, 2006): A filter method that aims to maximize the class-relevant information of the feature set.
- ICAP (Interaction Capping Criterion, Jakulin, 2005): A filter method that considers the complementary relationship between features.
- RFS (Robust Feature Selection, Nie et al., 2010): A method embedded in regression that uses joint $L^1$ and $L^2$ regularization of the weights.
- QS (Quick Selection, Atashgahi et al., 2022): A method embedded in sparse neural networks that combines denoising autoencoders and the $L^1$ norm of first layer neuron weights.
- STG (Stochastic Gates, Yamada et al., 2020): A method embedded in neural networks that controls the input layer neurons using a trainable probabilistic gate.
- LassoNet (Lemhadri et al., 2021): A method embedded in neural networks that adds a regularized residual connection from the input layer to the output. The residual connection controls the sizes of first layer weights.
- RigL (Evci et al., 2020): A method embedded in sparse neural networks that rotates features by pruning based on parameter weights and regrowing based on gradients. Feature selection can be performed by investigating first layer weights after training (Atashgahi et al., 2023).
- NeuroFS (Atashgahi et al., 2023): A method embedded in sparse neural networks that extends the ideas used in RigL to input neurons.

### A.2 RELCHANET SETUP

The parameters used for RelChaNet in the main experiment are as follows. We employ a single hidden layer neural network with 100 neurons and a ReLU activation function. For training, we use a batch size of 1024 and a learning rate of 0.001 for the Adam optimizer. If there are fewer cases in the dataset, full batches are used instead. The hyperparameters specific to our method are: $c_{\text{ratio}} = 0.2$ and $n_{\text{mb}} = 100$ for long datasets, and $c_{\text{ratio}} = 0.5$ and $n_{\text{mb}} = 5$ for wide datasets. Stopping is based on a combination of validation loss and the identified feature set. For this, the training data is split again into a training and a validation set. Training continues on the training set until the validation loss does not decrease for 100 input layer rotations or the set of $K$ features with the highest values in $\boldsymbol{h}$ remains unchanged for 100 rotations. Afterwards, the training is again performed on the complete training data for the determined number of rotations. For the flex algorithm, during this final training phase, the input layer is scaled from its initial size to the final size using a total of ten size change steps.

## B   DETAILED RESULTS

Table 2: Resulting accuracy of the studied methods for different numbers of selected features $K$ and datasets using the SVM downstream learner. Our proposed methods are "RCN" and "RCN flex". "All" is the accuracy using all features in the dataset. The best and second-best methods for each combination of $K$ and dataset are marked in bold and underlined, respectively. Entries represent the mean $\pm$ standard deviation of the downstream learner accuracy across five runs. Results for the baseline methods are reproduced from Atashgahi et al. (2023).

| | COIL-20 | HAR | ISOLET | MNIST | Fashion-MNIST | USPS | ARCENE | GLA-BRA-180 | Prostate-GE |
|---|---|---|---|---|---|---|---|---|---|
| All | 100.00 | 95.05 | 96.03 | 97.92 | 88.30 | 97.58 | 77.50 | 72.22 | 80.95 |
| **K = 25** | | | | | | | | | |
| NeuroFS | 95.86 ± 1.31 | 87.46 ± 0.79 | 86.22 ± 0.84 | 87.86 ± 1.77 | 79.38 ± 0.96 | 93.98 ± 0.87 | 63.00 ± 4.85 | 73.88 ± 3.80 | 88.58 ± 2.35 |
| LassoNet | 92.72 ± 0.85 | **93.00 ± 0.31** | 76.48 ± 0.39 | 86.40 ± 1.26 | 78.68 ± 0.55 | 94.04 ± 0.38 | 69.00 ± 2.55 | 76.12 ± 4.19 | 88.58 ± 2.35 |
| STG | 97.02 ± 1.41 | 87.48 ± 0.80 | 77.16 ± 4.34 | 85.24 ± 1.89 | 77.44 ± 0.53 | 94.04 ± 0.46 | 69.00 ± 5.15 | 67.22 ± 4.78 | 85.72 ± 3.00 |
| QS | 91.00 ± 4.21 | 87.14 ± 1.74 | 72.56 ± 6.53 | 85.25 ± 1.47 | 71.57 ± 1.97 | 93.00 ± 0.81 | 73.75 ± 8.20 | 69.45 ± 2.75 | 71.43 ± 12.16 |
| Fisher | 24.70 ± 0.00 | 77.10 ± 0.00 | 57.40 ± 0.00 | 74.40 ± 0.00 | 53.10 ± 0.00 | 82.00 ± 0.00 | 65.00 ± 0.00 | 58.30 ± 0.00 | **90.50 ± 0.00** |
| CIFE | 50.70 ± 0.00 | 80.20 ± 0.00 | 56.00 ± 0.00 | 80.90 ± 0.00 | 63.40 ± 0.00 | 50.20 ± 0.00 | 67.50 ± 0.00 | 61.10 ± 0.00 | 61.90 ± 0.00 |
| ICAP | 94.40 ± 0.00 | 84.50 ± 0.00 | 67.10 ± 0.00 | 81.60 ± 0.00 | 50.10 ± 0.00 | 89.90 ± 0.00 | 77.50 ± 0.00 | 69.40 ± 0.00 | 47.60 ± 0.00 |
| RFS | 88.20 ± 0.00 | 88.90 ± 0.00 | 76.50 ± 0.00 | - | - | 94.80 ± 0.00 | 77.50 ± 0.00 | - | 90.50 ± 0.00 |
| RigL | 92.38 ± 3.20 | 86.46 ± 1.47 | 79.98 ± 2.25 | 82.06 ± 0.99 | 74.12 ± 1.59 | 93.10 ± 0.62 | 74.50 ± 4.30 | 66.10 ± 3.22 | 78.08 ± 6.46 |
| RCN | 98.75 ± 0.31 | **92.07 ± 1.42** | **88.45 ± 1.16** | 93.04 ± 0.41 | **83.05 ± 0.40** | **95.82 ± 0.49** | 78.50 ± 6.52 | 75.00 ± 2.78 | 90.48 ± 0.00 |
| RCN flex | **98.89 ± 0.71** | 92.06 ± 0.97 | 88.28 ± 1.41 | **93.10 ± 0.25** | 82.70 ± 0.32 | 95.68 ± 0.08 | **80.50 ± 5.12** | **77.78 ± 1.96** | 88.57 ± 2.61 |
| **K = 50** | | | | | | | | | |
| NeuroFS | 98.78 ± 0.29 | 91.46 ± 0.72 | 92.62 ± 0.40 | 95.30 ± 0.41 | 83.78 ± 0.64 | 96.78 ± 0.17 | 76.50 ± 2.55 | **80.54 ± 4.96** | **90.50 ± 0.00** |
| LassoNet | 97.16 ± 1.06 | 93.74 ± 0.39 | 94.46 ± 0.21 | 82.58 ± 0.10 | 95.94 ± 0.15 | 71.00 ± 2.00 | 74.46 ± 4.78 | 88.58 ± 2.35 | |
| STG | 99.32 ± 0.40 | 91.22 ± 1.23 | 85.82 ± 2.83 | 93.20 ± 0.62 | 82.36 ± 0.52 | 96.62 ± 0.34 | 71.00 ± 2.55 | 70.00 ± 4.08 | 84.78 ± 3.55 |
| QS | 96.52 ± 1.53 | 91.96 ± 1.04 | 89.78 ± 1.80 | 93.62 ± 0.49 | 80.82 ± 0.51 | 95.52 ± 0.27 | 74.38 ± 4.80 | 72.20 ± 2.80 | 76.20 ± 7.53 |
| Fisher | 74.00 ± 0.00 | 79.80 ± 0.00 | 67.40 ± 0.00 | 81.90 ± 0.00 | 67.80 ± 0.00 | 91.00 ± 0.00 | 67.50 ± 0.00 | 63.90 ± 0.00 | 90.50 ± 0.00 |
| CIFE | 59.40 ± 0.00 | 84.20 ± 0.00 | 59.80 ± 0.00 | 89.30 ± 0.00 | 66.90 ± 0.00 | 61.30 ± 0.00 | 52.50 ± 0.00 | 58.30 ± 0.00 | 47.60 ± 0.00 |
| ICAP | 99.30 ± 0.00 | 88.70 ± 0.00 | 75.10 ± 0.00 | 89.00 ± 0.00 | 59.50 ± 0.00 | 95.20 ± 0.00 | 70.00 ± 0.00 | 72.20 ± 0.00 | 57.10 ± 0.00 |
| RFS | 95.80 ± 0.00 | **94.00 ± 0.00** | 91.50 ± 0.00 | - | - | 95.80 ± 0.00 | **77.50 ± 0.00** | - | 90.50 ± 0.00 |
| RigL | 97.86 ± 1.32 | 91.82 ± 0.30 | 89.58 ± 1.24 | 93.94 ± 0.63 | 81.92 ± 0.87 | 96.04 ± 0.58 | 77.00 ± 3.32 | 70.54 ± 4.16 | 79.06 ± 7.11 |
| RCN | **99.58 ± 0.29** | 93.74 ± 0.62 | 93.41 ± 0.25 | 96.69 ± 0.19 | **85.95 ± 0.22** | 96.83 ± 0.17 | 72.50 ± 5.59 | 73.33 ± 1.52 | 90.48 ± 0.00 |
| RCN flex | 99.51 ± 0.19 | 93.65 ± 0.36 | **93.46 ± 0.19** | **96.79 ± 0.11** | 85.84 ± 0.36 | **97.06 ± 0.23** | 76.00 ± 6.75 | 74.44 ± 2.32 | 89.52 ± 2.13 |
| **K = 75** | | | | | | | | | |
| NeuroFS | 99.06 ± 0.12 | 93.16 ± 0.79 | 94.04 ± 0.34 | 96.76 ± 0.22 | 85.70 ± 0.28 | 97.06 ± 0.15 | **82.00 ± 4.00** | **82.24 ± 3.31** | 89.54 ± 1.92 |
| LassoNet | 99.46 ± 0.35 | 94.62 ± 0.17 | 91.00 ± 0.62 | 96.00 ± 0.09 | 83.92 ± 0.13 | 96.36 ± 0.08 | 70.50 ± 2.45 | 76.64 ± 5.44 | **90.50 ± 0.00** |
| STG | 99.68 ± 0.22 | 92.42 ± 1.11 | 90.10 ± 2.17 | 95.52 ± 0.22 | 84.14 ± 0.43 | 96.88 ± 0.23 | 75.00 ± 2.74 | 71.08 ± 1.37 | 84.78 ± 3.55 |
| QS | 98.17 ± 1.16 | 93.50 ± 0.77 | 93.04 ± 0.46 | 95.98 ± 0.33 | 83.80 ± 0.53 | 96.85 ± 0.05 | 76.88 ± 2.72 | 73.60 ± 1.40 | 72.62 ± 9.78 |
| Fisher | 76.00 ± 0.00 | 81.70 ± 0.00 | 76.00 ± 0.00 | 87.10 ± 0.00 | 74.30 ± 0.00 | 94.40 ± 0.00 | 70.00 ± 0.00 | 66.70 ± 0.00 | **90.50 ± 0.00** |
| CIFE | 63.20 ± 0.00 | 84.80 ± 0.00 | 74.30 ± 0.00 | 92.70 ± 0.00 | 67.70 ± 0.00 | 68.00 ± 0.00 | 72.50 ± 0.00 | 58.30 ± 0.00 | 47.60 ± 0.00 |
| ICAP | 99.00 ± 0.00 | 89.20 ± 0.00 | 79.70 ± 0.00 | 92.40 ± 0.00 | 67.20 ± 0.00 | 95.30 ± 0.00 | 72.50 ± 0.00 | 72.20 ± 0.00 | 57.10 ± 0.00 |
| RFS | 99.70 ± 0.00 | 94.90 ± 0.00 | 93.90 ± 0.00 | - | - | **97.20 ± 0.00** | 80.00 ± 0.00 | - | 90.50 ± 0.00 |
| RigL | 99.20 ± 0.43 | 93.34 ± 0.47 | 92.32 ± 0.56 | 95.98 ± 0.51 | 84.52 ± 0.72 | 96.90 ± 0.24 | 81.50 ± 4.64 | 72.22 ± 4.98 | 79.06 ± 8.83 |
| RCN | **99.93 ± 0.16** | **95.31 ± 0.37** | 94.60 ± 0.49 | 97.49 ± 0.13 | 86.75 ± 0.25 | 97.15 ± 0.19 | 71.00 ± 7.42 | 77.78 ± 3.40 | 90.48 ± 0.00 |
| RCN flex | **99.93 ± 0.16** | 94.60 ± 0.65 | **94.88 ± 0.31** | **97.53 ± 0.11** | **86.76 ± 0.14** | 97.19 ± 0.10 | **82.00 ± 4.81** | 75.56 ± 3.04 | 90.48 ± 0.00 |
| **K = 100** | | | | | | | | | |
| NeuroFS | 99.18 ± 0.50 | 94.18 ± 0.29 | 95.06 ± 0.31 | 97.32 ± 0.17 | 86.64 ± 0.21 | 97.22 ± 0.12 | **82.00 ± 1.87** | **81.12 ± 2.05** | 89.54 ± 1.92 |
| LassoNet | 99.30 ± 0.00 | 95.14 ± 0.29 | 93.18 ± 0.22 | 96.64 ± 0.14 | 84.98 ± 0.18 | 97.04 ± 0.12 | 72.00 ± 4.30 | 79.46 ± 2.83 | **90.50 ± 0.00** |
| STG | 99.76 ± 0.12 | 92.82 ± 0.74 | 92.64 ± 0.56 | 96.38 ± 0.35 | 85.20 ± 0.58 | 97.08 ± 0.18 | 75.50 ± 3.67 | 72.20 ± 3.07 | 85.72 ± 3.00 |
| QS | 98.28 ± 1.15 | 94.06 ± 0.48 | 94.22 ± 0.28 | 96.85 ± 0.09 | 85.52 ± 0.15 | 97.00 ± 0.14 | 78.12 ± 1.08 | 73.60 ± 1.40 | 78.58 ± 9.82 |
| Fisher | 80.20 ± 0.00 | 83.80 ± 0.00 | 79.80 ± 0.00 | 90.70 ± 0.00 | 79.60 ± 0.00 | 96.50 ± 0.00 | 65.00 ± 0.00 | 66.70 ± 0.00 | **90.50 ± 0.00** |
| CIFE | 67.70 ± 0.00 | 85.30 ± 0.00 | 81.20 ± 0.00 | 95.10 ± 0.00 | 69.20 ± 0.00 | 78.00 ± 0.00 | 65.00 ± 0.00 | 58.30 ± 0.00 | 71.40 ± 0.00 |
| ICAP | **100.00 ± 0.00** | 92.10 ± 0.00 | 82.80 ± 0.00 | 95.00 ± 0.00 | 77.70 ± 0.00 | 95.40 ± 0.00 | **82.50 ± 0.00** | 69.40 ± 0.00 | 52.40 ± 0.00 |
| RFS | 100.00 ± 0.00 | 95.40 ± 0.00 | 94.40 ± 0.00 | - | - | **97.40 ± 0.00** | 80.00 ± 0.00 | - | 90.50 ± 0.00 |
| RigL | 99.40 ± 0.43 | 94.08 ± 0.26 | 93.66 ± 0.58 | 96.88 ± 0.22 | 85.82 ± 0.23 | 97.14 ± 0.10 | 80.00 ± 4.47 | 73.90 ± 3.76 | 81.92 ± 8.18 |
| RCN | 99.93 ± 0.16 | **95.61 ± 0.25** | **95.73 ± 0.46** | **97.80 ± 0.10** | **87.32 ± 0.15** | 97.34 ± 0.15 | 74.00 ± 2.85 | 77.22 ± 3.62 | 90.48 ± 0.00 |
| RCN flex | **100.00 ± 0.00** | 95.19 ± 0.19 | 95.21 ± 0.23 | 97.79 ± 0.07 | 87.21 ± 0.08 | 97.37 ± 0.13 | 77.50 ± 3.06 | 77.78 ± 4.39 | 90.48 ± 0.00 |

Table 3: Resulting accuracy of the studied methods for different downstream learners and datasets using $K = 50$ selected features. Our proposed methods are "RCN" and "RCN flex". "All" is the accuracy using all features in the dataset. The best and second-best methods for each combination of learner and dataset are marked in bold and underlined, respectively. Entries represent the mean $\pm$ standard deviation of the downstream learner accuracy across five runs. Results for the baseline methods are reproduced from Atashgahi et al. (2023).

| | COIL-20 | HAR | ISOLET | MNIST | Fashion-MNIST | USPS | ARCENE | GLA-BRA-180 | Prostate-GE |
|---|---|---|---|---|---|---|---|---|---|
| **Learner: ET** | | | | | | | | | |
| All | $100.00 \pm 0.00$ | $93.53 \pm 0.15$ | $94.05 \pm 0.32$ | $97.10 \pm 0.05$ | $87.19 \pm 0.13$ | $96.29 \pm 0.16$ | $79.50 \pm 4.85$ | $75.00 \pm 4.97$ | $88.57 \pm 3.81$ |
| NeuroFS | $99.94 \pm 0.12$ | $85.48 \pm 1.46$ | $91.46 \pm 0.73$ | $93.68 \pm 0.43$ | $84.26 \pm 0.55$ | $95.44 \pm 0.27$ | $75.00 \pm 5.24$ | $75.46 \pm 6.71$ | $\mathbf{90.50 \pm 0.00}$ |
| LassoNet | $99.76 \pm 0.12$ | $\mathbf{91.12 \pm 0.30}$ | $84.94 \pm 0.62$ | $92.96 \pm 0.15$ | $83.68 \pm 0.13$ | $94.86 \pm 0.22$ | $73.50 \pm 4.64$ | $\mathbf{76.12 \pm 3.80}$ | $89.54 \pm 1.92$ |
| STG | $\mathbf{100.00 \pm 0.00}$ | $88.68 \pm 0.42$ | $88.50 \pm 2.15$ | $90.38 \pm 0.42$ | $82.05 \pm 0.48$ | $94.32 \pm 0.21$ | $\underline{79.00 \pm 3.39}$ | $71.08 \pm 2.24$ | $83.84 \pm 3.80$ |
| QS | $99.25 \pm 0.47$ | $87.86 \pm 0.72$ | $88.78 \pm 1.86$ | $91.95 \pm 0.58$ | $81.28 \pm 0.54$ | $94.28 \pm 0.40$ | $73.75 \pm 4.15$ | $75.00 \pm 0.00$ | $77.38 \pm 5.19$ |
| Fisher | $96.86 \pm 0.43$ | $85.50 \pm 0.30$ | $81.42 \pm 0.59$ | $84.86 \pm 0.15$ | $72.06 \pm 0.08$ | $90.94 \pm 0.24$ | $60.00 \pm 1.58$ | $63.90 \pm 0.00$ | $\underline{90.50 \pm 0.00}$ |
| CIFE | $74.70 \pm 0.00$ | $85.30 \pm 0.00$ | $55.40 \pm 0.00$ | $87.60 \pm 0.00$ | $68.40 \pm 0.00$ | $82.70 \pm 0.00$ | $50.00 \pm 0.00$ | $69.40 \pm 0.00$ | $52.40 \pm 0.00$ |
| ICAP | $99.70 \pm 0.00$ | $89.20 \pm 0.00$ | $70.60 \pm 0.00$ | $87.80 \pm 0.00$ | $65.50 \pm 0.00$ | $93.50 \pm 0.00$ | $\mathbf{80.00 \pm 0.00}$ | $63.90 \pm 0.00$ | $81.00 \pm 0.00$ |
| RFS | $98.30 \pm 0.00$ | $89.70 \pm 0.00$ | $90.40 \pm 0.00$ | - | - | $94.70 \pm 0.00$ | $75.00 \pm 0.00$ | - | $90.50 \pm 0.00$ |
| RCN | $\underline{100.00 \pm 0.00}$ | $90.32 \pm 1.26$ | $\mathbf{92.65 \pm 0.52}$ | $\underline{95.30 \pm 0.12}$ | $\mathbf{85.70 \pm 0.22}$ | $\underline{95.76 \pm 0.13}$ | $72.50 \pm 9.35$ | $\underline{76.11 \pm 1.52}$ | $90.48 \pm 0.00$ |
| RCN flex | $100.00 \pm 0.00$ | $\underline{91.12 \pm 1.33}$ | $\underline{92.19 \pm 0.47}$ | $\mathbf{95.41 \pm 0.21}$ | $\underline{85.49 \pm 0.29}$ | $\mathbf{95.91 \pm 0.18}$ | $78.00 \pm 6.71$ | $75.00 \pm 3.40$ | $90.48 \pm 0.00$ |
| **Learner: KNN** | | | | | | | | | |
| All | $100.00$ | $87.85$ | $88.14$ | $96.91$ | $84.96$ | $97.37$ | $92.50$ | $69.44$ | $76.19$ |
| NeuroFS | $99.80 \pm 0.28$ | $84.64 \pm 1.77$ | $85.96 \pm 1.53$ | $91.64 \pm 0.57$ | $80.12 \pm 0.87$ | $96.18 \pm 0.49$ | $74.00 \pm 5.15$ | $64.42 \pm 5.38$ | $85.86 \pm 4.67$ |
| LassoNet | $98.84 \pm 0.20$ | $\underline{88.70 \pm 0.57}$ | $79.22 \pm 0.47$ | $91.38 \pm 0.36$ | $79.30 \pm 0.20$ | $95.70 \pm 0.26$ | $67.50 \pm 7.75$ | $\mathbf{68.90 \pm 4.07}$ | $82.86 \pm 3.80$ |
| STG | $\mathbf{99.94 \pm 0.12}$ | $87.86 \pm 0.39$ | $83.16 \pm 3.42$ | $87.16 \pm 0.64$ | $77.65 \pm 0.48$ | $95.14 \pm 0.45$ | $75.00 \pm 5.24$ | $58.90 \pm 7.52$ | $81.00 \pm 0.00$ |
| QS | $98.80 \pm 0.38$ | $85.88 \pm 1.13$ | $82.38 \pm 3.12$ | $89.30 \pm 0.76$ | $76.65 \pm 0.51$ | $95.17 \pm 0.45$ | $75.00 \pm 3.54$ | $\underline{66.70 \pm 0.00}$ | $65.47 \pm 8.37$ |
| Fisher | $95.80 \pm 0.00$ | $81.10 \pm 0.00$ | $74.10 \pm 0.00$ | $80.20 \pm 0.00$ | $63.70 \pm 0.00$ | $88.80 \pm 0.00$ | $70.00 \pm 0.00$ | $50.00 \pm 0.00$ | $85.70 \pm 0.00$ |
| CIFE | $71.20 \pm 0.00$ | $71.80 \pm 0.00$ | $44.60 \pm 0.00$ | $82.90 \pm 0.00$ | $61.60 \pm 0.00$ | $59.60 \pm 0.00$ | $70.00 \pm 0.00$ | $44.40 \pm 0.00$ | $57.10 \pm 0.00$ |
| ICAP | $98.60 \pm 0.00$ | $82.70 \pm 0.00$ | $59.00 \pm 0.00$ | $83.40 \pm 0.00$ | $59.30 \pm 0.00$ | $94.00 \pm 0.00$ | $65.00 \pm 0.00$ | $61.10 \pm 0.00$ | $66.70 \pm 0.00$ |
| RFS | $97.20 \pm 0.00$ | $\mathbf{90.30 \pm 0.00}$ | $\underline{87.20 \pm 0.00}$ | - | - | $95.40 \pm 0.00$ | $\mathbf{85.00 \pm 0.00}$ | - | $\mathbf{90.50 \pm 0.00}$ |
| RCN | $\underline{99.93 \pm 0.16}$ | $86.43 \pm 0.93$ | $\mathbf{88.21 \pm 0.46}$ | $\underline{94.48 \pm 0.20}$ | $\underline{82.01 \pm 0.20}$ | $\mathbf{96.65 \pm 0.20}$ | $73.00 \pm 7.37$ | $58.89 \pm 4.12$ | $87.62 \pm 2.61$ |
| RCN flex | $99.79 \pm 0.31$ | $86.40 \pm 1.14$ | $87.12 \pm 0.69$ | $\mathbf{94.62 \pm 0.24}$ | $\mathbf{82.10 \pm 0.56}$ | $\underline{96.48 \pm 0.39}$ | $76.50 \pm 2.24$ | $62.22 \pm 6.09$ | $\underline{89.52 \pm 2.13}$ |
| **Learner: SVM** | | | | | | | | | |
| All | $100.00$ | $95.05$ | $96.03$ | $97.92$ | $88.30$ | $97.58$ | $77.50$ | $72.22$ | $80.95$ |
| NeuroFS | $98.78 \pm 0.29$ | $91.46 \pm 0.72$ | $92.62 \pm 0.40$ | $95.30 \pm 0.41$ | $83.78 \pm 0.64$ | $96.78 \pm 0.17$ | $76.50 \pm 2.55$ | $\mathbf{80.54 \pm 4.96}$ | $\mathbf{90.50 \pm 0.00}$ |
| LassoNet | $97.16 \pm 1.06$ | $\underline{93.74 \pm 0.39}$ | $84.90 \pm 0.22$ | $94.46 \pm 0.21$ | $82.58 \pm 0.10$ | $95.94 \pm 0.15$ | $71.00 \pm 2.00$ | $\underline{74.46 \pm 4.78}$ | $88.58 \pm 2.35$ |
| STG | $99.32 \pm 0.40$ | $91.22 \pm 1.23$ | $85.82 \pm 2.83$ | $93.20 \pm 0.62$ | $82.36 \pm 0.52$ | $96.62 \pm 0.34$ | $71.00 \pm 2.55$ | $70.00 \pm 4.08$ | $84.78 \pm 3.55$ |
| QS | $96.52 \pm 1.53$ | $91.96 \pm 1.04$ | $89.78 \pm 1.80$ | $93.62 \pm 0.49$ | $80.82 \pm 0.51$ | $95.52 \pm 0.27$ | $74.38 \pm 4.80$ | $72.20 \pm 2.80$ | $76.20 \pm 7.53$ |
| Fisher | $74.00 \pm 0.00$ | $79.80 \pm 0.00$ | $67.40 \pm 0.00$ | $81.90 \pm 0.00$ | $67.80 \pm 0.00$ | $91.00 \pm 0.00$ | $67.50 \pm 0.00$ | $63.90 \pm 0.00$ | $\underline{90.50 \pm 0.00}$ |
| CIFE | $59.40 \pm 0.00$ | $84.20 \pm 0.00$ | $59.80 \pm 0.00$ | $89.30 \pm 0.00$ | $66.90 \pm 0.00$ | $61.30 \pm 0.00$ | $52.50 \pm 0.00$ | $58.30 \pm 0.00$ | $47.60 \pm 0.00$ |
| ICAP | $99.30 \pm 0.00$ | $88.70 \pm 0.00$ | $75.10 \pm 0.00$ | $89.00 \pm 0.00$ | $59.50 \pm 0.00$ | $95.20 \pm 0.00$ | $70.00 \pm 0.00$ | $72.20 \pm 0.00$ | $57.10 \pm 0.00$ |
| RFS | $95.80 \pm 0.00$ | $\mathbf{94.00 \pm 0.00}$ | $91.50 \pm 0.00$ | - | - | $95.80 \pm 0.00$ | $\mathbf{77.50 \pm 0.00}$ | - | $90.50 \pm 0.00$ |
| RigL | $97.86 \pm 1.32$ | $91.82 \pm 0.30$ | $89.58 \pm 1.24$ | $93.94 \pm 0.63$ | $81.92 \pm 0.87$ | $96.04 \pm 0.58$ | $\underline{77.00 \pm 3.32}$ | $70.54 \pm 4.16$ | $79.06 \pm 7.11$ |
| RCN | $\mathbf{99.58 \pm 0.29}$ | $93.74 \pm 0.62$ | $\underline{93.41 \pm 0.25}$ | $\underline{96.69 \pm 0.19}$ | $\mathbf{85.95 \pm 0.22}$ | $\underline{96.83 \pm 0.17}$ | $72.50 \pm 5.59$ | $73.33 \pm 1.52$ | $90.48 \pm 0.00$ |
| RCN flex | $\underline{99.51 \pm 0.19}$ | $93.65 \pm 0.36$ | $\mathbf{93.46 \pm 0.19}$ | $\mathbf{96.79 \pm 0.11}$ | $\underline{85.84 \pm 0.36}$ | $\mathbf{97.06 \pm 0.23}$ | $76.00 \pm 6.75$ | $74.44 \pm 2.32$ | $89.52 \pm 2.13$ |

## C  AUXILARY EXPERIMENT

To complement the experiments in Section 4, we test the performance of RelChaNet against two baseline methods on four additional datasets. CIFAR-10 and CIFAR-100 are two additional long datasets representing complex prediction tasks, while BASEHOCK and SMK are two additional wide datasets. Their dimensions are shown in Table 4. Each condition was run five times, except for NeuroFS on the CIFAR datasets, where we limited runs to a single iteration to ensure that the runtime remained below 12 hours per condition.

Table 4: Dataset dimensions and domain

|  | Cases | Features | Domain | Reference |
|---|---|---|---|---|
| CIFAR-10 | 60000 | 3072 | Image | Krizhevsky (2009) |
| CIFAR-100 | 60000 | 3072 | Image | Krizhevsky (2009) |
| BASEHOCK | 1993 | 4862 | Text | Lang (1995) |
| SMK | 187 | 19993 | Genomics | Spira et al. (2007) |

The results in Table 5 show that RCN and RCN flex perform best for the CIFAR-10 dataset, indicating their potential for more complex datasets. However, they are outperformed by NeuroFS for CIFAR-100, potentially due to their much smaller architecture. For the two wide datasets, the results are mixed: RCN and RCN flex outperform the other approaches on SMK but fall behind on BASEHOCK.

Table 5: Resulting accuracy of the studied methods for different numbers of selected features $K$ and datasets using the SVM downstream learner. Our proposed methods are "RCN" and "RCN flex". "All" is the accuracy using all features in the dataset. The best and second-best methods for each combination of $K$ and dataset are marked in bold and underlined, respectively. Entries represent the mean $\pm$ standard deviation of the downstream learner accuracy across five runs, except for the CIFAR datasets and NeuroFS method, which use a single run.

|  | CIFAR-10 | CIFAR-100 | BASEHOCK | SMK |
|---|---|---|---|---|
| All | 54.36 | 26.39 | 94.24 | 84.21 |
| **Average** |  |  |  |  |
| NeuroFS | 46.62 | **20.95** | 89.00 | 79.20 |
| LassoNet | 28.60 | 11.45 | **91.44** | 78.03 |
| RCN | 46.80 | 19.26 | 85.87 | **82.76** |
| RCN flex | **47.36** | 19.77 | 85.14 | 82.11 |
| **K = 25** |  |  |  |  |
| NeuroFS | 40.40 | **17.40** | 85.46 ± 2.10 | 77.34 ± 5.76 |
| LassoNet | 23.30 ± 0.96 | 9.58 ± 1.05 | **89.82 ± 1.21** | 74.74 ± 3.00 |
| RCN | 40.71 ± 1.75 | 15.33 ± 0.84 | 82.31 ± 1.82 | **82.63 ± 6.06** |
| RCN flex | **41.82 ± 0.64** | 15.26 ± 1.09 | 81.40 ± 0.88 | 77.89 ± 6.34 |
| **K = 50** |  |  |  |  |
| NeuroFS | 46.30 | **21.10** | 88.08 ± 0.70 | 81.56 ± 2.65 |
| LassoNet | 28.77 ± 5.00 | 10.55 ± 0.50 | **91.98 ± 1.16** | 80.53 ± 3.99 |
| RCN | 46.65 ± 0.60 | 18.98 ± 0.90 | 86.47 ± 1.45 | **83.68 ± 4.32** |
| RCN flex | **47.23 ± 0.87** | 19.13 ± 1.34 | 84.56 ± 1.67 | 82.11 ± 3.43 |
| **K = 75** |  |  |  |  |
| NeuroFS | **50.30** | **22.10** | 90.86 ± 2.20 | 78.40 ± 3.89 |
| LassoNet | 30.22 ± 1.54 | 12.41 ± 2.15 | **91.88 ± 1.01** | 78.42 ± 7.54 |
| RCN | 49.28 ± 0.47 | 20.40 ± 0.63 | 87.47 ± 1.59 | 82.63 ± 2.35 |
| RCN flex | 49.65 ± 0.38 | 21.52 ± 0.58 | 86.87 ± 1.69 | **83.16 ± 3.53** |
| **K = 100** |  |  |  |  |
| NeuroFS | 49.50 | **23.20** | 91.62 ± 2.08 | 79.48 ± 5.69 |
| LassoNet | 32.12 ± 0.56 | 13.25 ± 2.22 | **92.08 ± 0.52** | 78.42 ± 2.20 |
| RCN | 50.58 ± 0.40 | 22.34 ± 0.43 | 87.22 ± 1.42 | 82.11 ± 2.88 |
| RCN flex | **50.73 ± 0.34** | 23.19 ± 0.18 | 87.72 ± 2.56 | **85.26 ± 1.44** |

