# OpenReview forum: "RelChaNet: Neural Network Feature Selection using Relative Change Scores"
_ICLR.cc/2025/Conference — Submitted to ICLR 2025_

### Official Review · Reviewer_JgCy · 2024-11-02

**Soundness:** 2
**Presentation:** 2
**Contribution:** 2
**Rating:** 5
**Confidence:** 5

**Summary:**

The primary focus of this paper is on feature selection algorithms in neural networks. Thus they  introduce RelChaNet, a feature selection algorithm that uses neuron pruning and regrowth in the input layer of a dense neural network.

**Strengths:**

The method proposed in this paper is very simple and easy to implement.

**Weaknesses:**

1、The solutions in this paper are almost identical to methods like dropout, pruning, and regularization in neural networks to prevent overfitting, making it difficult to identify the novelty of the proposed approach.
2、The effectiveness of the proposed method is also not better than the state-of-the-art (SOTA) results.
3、The threshold C_ratio  in the feature selection algorithm lacks theoretical guidance or a defined method for setting it.
4、The quality of English writing in the paper needs improvement.
5、The paper lacks an evaluation and summary of related work, as well as an explanation of the challenges present in the problem.

**Questions:**

N/A

---

> ### Author Response · Authors · 2024-11-25
> **Response to Reviewer JgCy02**
>
> We greatly appreciate your constructive feedback and thoughtful comments. We have addressed each of your points in detail below, and have updated the manuscript accordingly (highlighted in cyan).
>
> ---
>
> ### [W1] Novelty of RelChaNet
> Thanks for bringing up the question of novelty. RelChaNet’s key novelties are firstly combining dense neural networks with pruning and regrowth from the sparse neural network literature, and secondly, leveraging theoretical advantages of a novel combination of relative change based pruning and random regrowth.
>
> While our approach does not make use of dropout, it does make use of pruning as a form of regularization. There is no consensus yet on how to best perform pruning in neural networks, as various ideas have evolved into a vast array of literature where specific implementations are recognized as novel [1]. In line with this, our approach represents a specific pruning and regrowth protocol tailored for the task of feature selection.
>
> [1] Cheng, Hongrong, Miao Zhang, and Javen Qinfeng Shi. “A Survey on Deep Neural Network Pruning: Taxonomy, Comparison, Analysis, and Recommendations.” IEEE Transactions on Pattern Analysis and Machine Intelligence 46, no. 12 (December 2024): 10558–78. https://doi.org/10.1109/TPAMI.2024.3447085.
>
> ---
>
> ### [W2] Performance comparison to state-of-the-art results
> We appreciate discussing the performance of RelChaNet in comparison to previously suggested methods for supervised feature selection. In our experiments, RelChaNet outperforms state-of-the-art methods for the six “long” datasets and demonstrates comparable performance on three “wide” datasets. We are very open to suggestions for including additional datasets or baseline methods to bolster our evaluation.
>
> ---
>
>
> ### [W3] Setting the $c_{\text{ratio}}$ hyperparameter
> Thanks for highlighting the setting of the candidate ratio hyparameter in the RelChaNet algorithm. In our updated paper, we updated our theoretical treatment of the parameter in the context of discussing random regrowth in Section 3. Accordingly, the role of $c_{\text{ratio}}$ is to set the size of the input layer in an exploration vs. exploitation tradeoff: A small input layer may hinder the ability of the random regrowth to find relevant feature sets while a large input layer implies an increase in noise.
>
> Regarding methods for setting the $c_{\text{ratio}}$ parameter, a practical starting point is to use one of the two configurations we used in our experiments, depending on whether the dataset is long or wide. Since hyperparameters can be further optimized for specific datasets, we provide an orientation for feasible ranges for both $c_{\text{ratio}}$ and $n_{\text{mb}}$ in an analysis across ~50 configurations in Section 4.2.
>
> ---
>
> ### [W4] Quality of English writing
> Thank you for your feedback on the writing quality. We are eager to ensure that the writing does not stand in the way of the content. With this in mind, we have edited expressions throughout the paper to increase clarity and flow, and we specifically reworked Section 3 to make it easier to read. We remain open to further pointers on parts that need improvement in writing quality.
>
> ---
>
> ### [W5] Related work and problem introduction
> Thanks for highlighting the importance of reviewing related work. We include an evaluation and summary of related neural network feature selection approaches in Section 2. We are of course open to further suggestions for improving this summary.
>
> Thank you for pointing out the need to highlight the specific challenges in feature selection for neural networks. In our updated paper, we have added an explicit discussion following the definition of the feature selection task. We emphasize that the key challenge lies in implementing effective $L^0$ regularization, as exact solutions are computationally prohibitive and become intractable in high-dimensional settings.
>
> ---
>
> Thank you for your help in improving our work. We hope that these updates address your concerns and are looking forward to your feedback.

---

### Official Review · Reviewer_HsnL · 2024-11-03

**Soundness:** 2
**Presentation:** 3
**Contribution:** 2
**Rating:** 5
**Confidence:** 4

**Summary:**

This paper introduces a novel feature selection method designed for dense neural networks. RelChaNet leverages neuron pruning and random regrowth at the input layer, selecting features based on a relative change score calculated from gradient sums over multiple mini-batches. Experiments across nine datasets demonstrate that RelChaNet generally outperforms existing feature selection techniques, particularly enhancing accuracy by 2% on MNIST. The paper also introduces an adaptive extension, “RelChaNet flex,” which adjusts the input layer size dynamically based on validation loss trends.

**Strengths:**

The paper presents extensive results across diverse datasets, showing superior performance over baseline feature selection methods and emphasizing improvements in interpretability and computational efficiency.

**Weaknesses:**

- The applicability of this method is uncertain. In some cases, neurons may exhibit monosemanticity (e.g., in a neural network performing simple arithmetic tasks, where each neuron has a clear, isolated role). However, in other cases, groups of neurons may collectively capture shared or complex features. This method seems most effective when monosemanticity is prevalent in the dataset, and it may struggle with datasets that contain intricate concepts requiring shared neuron activation.
- The experiments focus primarily on datasets with more cases than features (“long” datasets). To strengthen the evaluation, RelChaNet should be tested on additional “wide” datasets to assess its performance on high-dimensional data. Additionally, the current experiments use relatively simple datasets. Expanding the evaluation to include more complex datasets, such as ImageNet, would help demonstrate the method’s robustness in handling challenging data.
- The paper lacks a theoretical explanation for the random neuron regrowth process. Without a clear rationale, the consistency and predictability of the feature selection results may be affected.

**Questions:**

- How does RelChaNet perform on very high-dimensional data with more features than samples? Although the algorithm shows effectiveness on “long” datasets, further validation on “wide” datasets would provide a more complete view of its generalizability.
- What impact does the randomness in neuron regrowth have on feature selection stability? Since neurons are randomly regrown, it would be useful to understand how this randomness affects the repeatability of selected features and model accuracy.
- How does the algorithm’s computational efficiency compare with other pruning-based feature selection methods? Given its relative complexity, a comparison of runtime across similar algorithms would be helpful in evaluating RelChaNet’s scalability.
- Could hyperparameters like cratio and nmb be optimized for specific types of datasets? Insights into parameter tuning would provide valuable guidance for applying RelChaNet in various contexts, especially for practitioners without prior knowledge of optimal settings.

---

> ### Author Response · Authors · 2024-11-25
> **Response to Reviewer vHsnL (1/2)**
>
> We greatly appreciate your constructive feedback and thoughtful comments. We have addressed each of your points in detail below, and have updated the manuscript accordingly (highlighted in cyan).
>
> ---
>
> ### [W1] Making use of feature interactions
> Thank you for bringing up the discussion on monosemanticity. In our understanding, this corresponds to the presence of interactions between features when referring to monosemanticity in input neurons. We see both theoretical potential and empirical evidence that RelChaNet does make use of interactions between features. The theoretical potential lies in the fact that it uses a neural network architecture that can generally make use of interactions and is not significantly hindered by the applied pruning and regrowth protocol. With the caveat that interacting features need to be in the input layer at the same time, our approach can identify them using the relative change score.
>
> Empirically, RelChaNet demonstrates its use of feature interactions by outperforming methods that are more limited to monosemanticity, such as the Fisher Score, as well as methods capable of utilizing complex patterns (e.g. LassoNet, NeuroFS). We bolstered this observation in our additional experiments with the CIFAR-10 and CIFAR-100 datasets, where less monosemanticity is present. In these experiments, RelChaNet shows overall good performance and significantly outperforms LassoNet (see Appendix C).
>
> ---
>
> ### [W2] Additional datasets
> Thanks for the suggestion for adding more datasets to strengthen the experimental evaluation. We have added evaluations on four additional datasets in an auxiliary analysis (see Appendix C). This includes two additional wide datasets (BASEHOCK and SMK) as well as two more complex long datasets (CIFAR-10 and CIFAR-100). Respecting computational constraints, we limited the analysis to the most relevant alternatives, LassoNet and NeuroFS, and conducted only one run per condition for NeuroFS to keep the analysis under 12 hours per condition. Despite this limitation, we believe these results still provide valuable insights.
>
> The results for the wide datasets are in line with the observations of the main experiment: RelChanet outperforms baseline methods on SMK but not on BASEHOCK. For the more complex datasets, RelChaNet demonstrates competitive performance, outperforming both alternatives on CIFAR-10. On CIFAR-100, while RelChaNet significantly outperforms LassoNet, NeuroFS exhibits slightly superior performance.
>
> We hope this additional evidence demonstrates the robustness of RelChaNet in various scenarios, and we look forward to your feedback on these additions.
>
> ---
>
> ### [W3] Random regrowth rationale
> Thanks for pointing this out. In a dedicated paragraph in Section 3 of the updated paper, we elaborate more clearly on the rationale of the random regrowth and the involved exploration exploitation tradeoff. The key points are:
>
> - The key motivation for random regrowth lies in giving candidate features multiple mini-batches to demonstrate their relevance instead of selecting candidates based on a metric prior to inclusion as candidates. This facilitates the identification of features that do not have a straightforward relationship with the output but contribute to complex patterns that only emerge over time.
> - The main downside is that it may take many rotations until sets of features that interact are included together in the input layer. This can however by counteracted by increasing the $c_{\text{ratio}}$ hyperparameter.
> - Increasing the  $c_{\text{ratio}}$ parameter in turn introduces more noise into the network by enlarging the input layer. As this noise may disrupting the learning process,  the  $c_{\text{ratio}}$ parameter represents a exploration and exploitation tradeoff
>
> ---
>
> ### [Q1] Additional wide datasets
> Thank you for suggesting the inclusion of additional high-dimensional datasets. We have implemented this and report the results in our response to [W2]. The findings are consistent with those of our main experiment.

---

> ### Author Response · Authors · 2024-11-25
> **Response to Reviewer vHsnL (2/2)**
>
> ### [Q2] Random regrowth and feature selection stability
> Thank you for raising this very interesting question about feature selection stability, which we investigated in a small experiment detailed below. In summary, when we reduce the impact of random regrowth by increasing the $c_{\text{ratio}}$ parameter, we observe an increase in feature selection stability. However, both the RCN model accuracy and downstream SVM accuracy decrease and get less stable. This provides evidence for the exploration/exploitation tradeoff we mention in our response to [W3]: Due to the higher amount of noise in the larger input layer, the RCN is hindered in honing in on good solutions. In effect, we either find slightly worse sets that are overlapping between runs or better sets that have less overlap.
>
> To study the impact of random regrowth, we varied the $c_{\text{ratio}}$ parameter. With a smaller input layer, random regrowth may impact stability more since features are not guaranteed to appear together. With a large input layer (e.g. $c_{\text{ratio}}$ = .8), random regrowth has less effect. We confirmed this in a small experiment where we measured the Jaccard indices for feature set overlap [1]. We used the MNIST dataset, K=50 selected features, three $c_{\text{ratio}}$ values, and 20 runs each. We found that increasing $c_{\text{ratio}}$ leads to greater overlap in selected feature sets, decreased RCN and SVM accuracies, and increased variance in accuracy. We observed similar trends in three other datasets.
>
> | $c_{\text{ratio}}$ | Jaccard | RCN Accuracy | SD   | SVM Accuracy | SD   |
> |--------------------|---------|--------------|------|--------------|------|
> | 0.2                | 0.13    | 95.80        | 0.43 | 96.84        | 0.16 |
> | 0.5                | 0.15    | 93.95        | 1.05 | 96.71        | 0.14 |
> | 0.8                | 0.21    | 91.50        | 1.87 | 96.06        | 0.22 |
>
> [1] Khaire, Utkarsh Mahadeo, and R. Dhanalakshmi. “Stability of Feature Selection Algorithm: A Review.” Journal of King Saud University - Computer and Information Sciences 34, no. 4 (April 2022): 1060–73. https://doi.org/10.1016/j.jksuci.2019.06.012.
>
> ---
>
> ### [Q3] Computational efficiency among pruning-based methods
> Thanks for highlighting the importance of comparing computational efficiency. Currently, we are aware of only one other pruning-based feature selection method, NeuroFS, which we have included in our efficiency comparison in Section 4.2. In this comparison, RelChaNet significantly outperforms NeuroFS.
>
> ---
>
> ### [Q4] Hyperparameter optimization
> Thanks for bringing up hyperparameter optimization for specific datasets. In our preliminary testing, we identified two configurations that perform well for wide and long datasets, respectively, and used these throughout the experiments ($c_{\text{ratio}}$ = .2, $n_{\text{mb}}$ = 100 for long, $c_{\text{ratio}}$ =.5, $n_{\text{mb}}$  = 5 for wide datasets).
>
> For specific datasets, there are likely hyperparameter sets that work even better. We provide some further insight into parameter tuning in an auxiliary analysis in Section 4 where we report the performance of ~50 different configurations for one wide and one long dataset each.
>
> A useful takeaway for practitioners is that for long datasets, there is a broader range of effective hyperparameter sets compared to wide datasets. These ranges can serve as a starting point for tuning. We are, of course, open to suggestions to further enrich this analysis.
>
> ---
>
> Thank you for your help in improving our work. We hope that these updates address your concerns and are looking forward to your feedback.

---

### Official Review · Reviewer_CZGE · 2024-11-04

**Soundness:** 3
**Presentation:** 2
**Contribution:** 2
**Rating:** 6
**Confidence:** 4

**Summary:**

This paper proposes a new algorithm for feature selection using Multi Layer Perceptron and a prune and regrowth strategy for the neurons from the input layer. The empirical evaluation shows that the proposed method outperforms several state-of-the-art baselines in a substantial number of scenarios.

**Strengths:**

S1) The proposed method is novel.

S2) Random neuron regrowth likely helps reduce bias in the final ranking of input features.

S3) The proposed method obtains a beneficial performance improvement on top of the state-of-the-art as illustrated on several datasets.

**Weaknesses:**

W1) The paper is somewhat difficult to read. Particularly, Section 3 is a bit hard to read due to too much text and details.

W2) "The paper needs careful proofreading. Some statements are unclear or inaccurately phrased. E.g., lines 44-45, Mocanu et al. 2018, Evci et al. 2020, employed connections pruning and regrow directly, while neuron pruning, and regrowth become rather an indirect output; lines 124 -> this is rather structured sparsity

**Questions:**

Q1) Can you try to enhance Section 3 by describing it with a proper mathematical formalism?

Q2) Which is better the main algorithm proposed in Section 3 or its extension from Section 3.1? Proposing two new algorithms which perform relatively similar is confusing.

Q3) Is it the case that the proposed approach underperforms on the widest dataset (about 50k features) because the random growth needs more training epochs to explore this very large search space? Did you try to train longer in a systematic manner for this dataset? Perhaps by creating an artificial dataset you may be able to perform a more granular analysis on how well the proposed method scales with the number of features and samples?

Q4) The computational analyze from Section 4.2 seems a bit forced. Probably, it would be fairer to try using relatively similar network sizes for all methods and report of course also their accuracies. Also, the sparse networks are really sparse or simulated with binary masks?

Q5) As far I was able to understand the work is about supervised feature selection. Can you please clarify?

---

> ### Author Response · Authors · 2024-11-25
> **Response to Reviewer CZGE (1/2)**
>
> We greatly appreciate your constructive feedback and thoughtful comments. We have addressed each of your points in detail below, and have updated the manuscript accordingly (highlighted in cyan).
>
> ---
>
> ### [W1] Improving readability
> Thank you for your feedback on readability, which motivated us to make several improvements. To improve the readability of Section 3 specifically, we named the paragraphs, cut redundant information between text and algorithm as well as some less important details. Additionally, we reworked the wording throughout the paper to enhance clarity and flow. For example, we added signposting to the additional analysis in Section 4.2. Please also see the updated manuscript for all changes. We hope these changes address your concerns and make our paper more accessible.
>
> ---
>
> ### [W2] Improving clarity and phrasing
> Thank you for highlighting the need for improved clarity and accuracy in certain statements. We went over the paper to improve clarity in statements, including the two you pointed to. Here are some examples but please also see the updated paper with highlighted changes:
>
> - We simplified the sentence around lines 44-45 to “Recently, it was shown that sparse neural network training (Mocanu et al., 2018; Evci et al., 2020) can be adapted to achieve a dominant feature selection performance (Liu et al., 2024; Atashgahi et al., 2024; Sokar et al., 2024).”
> - Later in the paper, we clarified that Mocanu et al. and Evci et al. perform weight pruning, while Atashgahi et al. extends this to include input layer neuron pruning (lines 135-137).
> - We added that neuron pruning is a case of structured sparsity  “One method to achieve this is structured sparsity, such as neuron pruning, where all of a neuron's outgoing weights are set to 0.” (l.124)
> - We specified that Molchanov is a neuron/filter pruning method (l.127)
> - We clarified the description of how NeuroFS regrows neurons. (l.139)
>
> ---
>
>
> ### [Q1] Additional mathematical formalism
> Thank you for the suggestion. To enhance the formal description of our approach in Algorithm 1, we have also expressed the input layer rotation more formally. We provide accompanying remarks in the main text to complement the formal description in the algorithm. We hope these changes address your concern.
>
> ---
>
> ### [Q2] Two RelChaNet versions
> Thank you for highlighting this point. Our decision to include both algorithms is based on their distinct contributions, which we have clarified in the updated paper.
>
> RelChaNet flex performs better on one dataset and is theoretically significant because the adaptive adjustment of the input layer size navigates an exploration vs. exploitation tradeoff (see the updated paper at lines 238-240). Moreover, an adaptive input layer aligns with previous work, which utlized shrinking input layers during training [1].
>
> The base algorithm, on the other hand, stands out for its simplicity and competitive performance in most scenarios, which we believe is worth reporting. We have updated the manuscript to include this reasoning in the Discussion (lines 482-483). We hope this addresses your concern and conveys to the reader why we opted for including both versions.
>
> [1] Atashgahi, Zahra, Xuhao Zhang, Neil Kichler, Shiwei Liu, Lu Yin, Mykola Pechenizkiy, Raymond Veldhuis, and Decebal Constantin Mocanu. “Supervised Feature Selection with Neuron Evolution in Sparse Neural Networks.” Transactions on Machine Learning Research, 2023. https://openreview.net/forum?id=GcO6ugrLKp.

---

> ### Author Response · Authors · 2024-11-25
> **Response to Reviewer CZGE (2/2)**
>
> ### [Q3] Performance on GLA-BRA-180
> Thanks for the suggestion to look into the underperformance in the widest dataset (GLA-BRA-180) and try increasing training duration. With our combined validation accuracy and feature set stability stopping rule the RCN method took on average 478 epochs in the experiment for K=50 selected features. Following your suggestion, we increased the training epochs by factors of 2, 5, and 10:
>
> | Epochs | Accuracy | SD   |
> |--------|----------|------|
> | 478    | 73.33    | 1.52 |
> | 954    | 77.22    | 4.10 |
> | 2389   | 76.67    | 3.97 |
> | 4779   | 69.72    | 4.62 |
>
> The results indeed show that training for the double number of epochs puts our methods much closer to the state of the art performance (80.54). This effect drops off with longer training, presumably through overfitting.
>
> From these results, we conclude that our stopping protocol could be improved in general, or adjusted individually for each dataset, possibly using validation data. Note that large feature spaces do not generally pose a problem for our current training protocol, as evidenced by the favorable results for a dataset with 20k features (see Appendix C).
>
> We appreciate the suggestion to test increasingly large artificial datasets, as it could provide valuable insights into the scalability of our method. However, we are unsure how to best prioritize this investigation, particularly as further exploration of the tuning protocol now appears promising. Thank you again for this insightful suggestion, which has highlighted potential areas for improvement.
>
> ---
>
> ### [Q4] Computational efficiency analysis
> Thanks for sharing the concern and the suggestion of extending this analysis with more similar architectures. The goal of our computational efficiency analysis is to compare versions that are most relevant for feature selection applications, either through being among the ones tested in literature experiments (NeuroFS) or by being the package default (LassoNet). For LassoNet, we opted for the package default instead of a paper architecture because it is the smaller of both and matches our own.
>
> We currently consider extending this analysis a low priority, as comparing more similar architectures would involve less relevant architectures from a feature selection perspective. For example, we would not recommend using the large architecture in NeuroFS for RelChaNet because overfitting would likely obstruct gains in feature selection performance. A similar investigation has already been made for LassoNet [1]. The NeuroFS authors reported that using their larger architecture for LassoNet lead to permissive increases in runtime in some scenarios and no significant benefit in the feature selection performance otherwise. Similarly, using a small architecture for NeuroFS would likely negate the beneficial effects of sparsity, making such a configuration less relevant for feature selection.
>
> Thanks for bringing up the topic of simulated sparsity. To our knowledge, the sparse networks used in NeuroFS employ binary masks rather than a purely sparse implementation, impacting efficiency. We added this to the report as a caveat: “However, NeuroFS utilizes binary masks to implement sparse networks, and future advancements in hardware optimized for sparse matrix computations could improve its efficiency.”(l.412)
>
> We hope this clarification of the focus of our analysis addresses your concern and we are, of course, open for further discussion.
>
> [1] Atashgahi, Zahra, Xuhao Zhang, Neil Kichler, Shiwei Liu, Lu Yin, Mykola Pechenizkiy, Raymond Veldhuis, and Decebal Constantin Mocanu. “Supervised Feature Selection with Neuron Evolution in Sparse Neural Networks.” Transactions on Machine Learning Research, 2023. https://openreview.net/forum?id=GcO6ugrLKp.
>
> ---
>
> ### [Q5] Supervised feature selection
> Yes, RelChaNet is a supervised feature selection method. We added this for clarity in the abstract and in the main text at the beginning of Section 3. Thanks for pointing this out.
>
> ---
>
> Thank you for your help in improving our work. We hope that these updates address your concerns and are looking forward to your feedback.

---

> ### Comment · Reviewer_CZGE · 2024-12-01
> **Rebuttal acknowledged**
>
> I thank the authors for trying to address a majority of my concerns. After considering their answers and the new experiments, I am inclined to increase my rating from 5 to 6.

---

### Official Review · Reviewer_vxF4 · 2024-11-04

**Soundness:** 2
**Presentation:** 3
**Contribution:** 1
**Rating:** 5
**Confidence:** 3

**Summary:**

The paper introduces RelChaNet, a novel feature selection algorithm leveraging neural networks. Key innovations include neuron pruning and regrowth mechanisms focused on the input layer. The pruning process uses a "relative change score", measuring the impact each feature has on the network's structure and function after its inclusion. Unique to RelChaNet is the flexibility to adapt input layer size dynamically during runtime, enhancing the algorithm's adaptability to varied datasets.

The method was benchmarked against other state-of-the-art feature selection algorithms on nine datasets, showing superior performance in terms of predictive accuracy, especially on datasets with more samples than features, achieving a 2% improvement on MNIST. However, RelChaNet exhibits comparable performance on datasets with more features than samples. Notably, it also offers competitive computational efficiency, making it a robust alternative for neural network-based feature selection tasks

**Strengths:**

- Given the ever-increasing computational demand in the deep learning field, RelChaNet addresses the critical need to reduce this load by proposing a novel deep learning feature selection method.
- The authors conduct experiments across a broad range of competing feature selection methods and a diverse set of data domains.
- RelChaNet (flex) demonstrates strong performance and robustness by outperforming other evaluated methods on 7 out of 9 datasets.

**Weaknesses:**

The primary weakness of this paper, as I see it, is that it evaluates RelChaNet on datasets that do not intrinsically demand the non-linear feature selection capabilities that deep learning methods like RelChaNet are designed to offer. For example, MNIST is a well-understood dataset where simpler, linear methods often perform exceptionally well. Linear methods like PCA, for instance, achieve 98.0% accuracy on MNIST with K=25 features when using the SVC downstream learner, notably outperforming RelChaNet’s ~93% accuracy. This raises questions about whether RelChaNet’s deep learning-based approach is meaningful for such datasets and whether it would generalize well to more complex, non-linear datasets (e.g., CIFAR-10, Imagenet).

To demonstrate the effectiveness of RelChaNet, the evaluation should focus on datasets with complex, non-linear relationships where simpler methods struggle.

**Questions:**

How do RelChaNet and other competing feature selection methods perform on complex datasets such as CIFAR-10, CIFAR-100, and Imagenet?

---

> ### Author Response · Authors · 2024-11-25
> **Response to Reviewer vxF4**
>
> We greatly appreciate your constructive feedback and thoughtful comments. We have addressed each of your points in detail below, and have updated the manuscript accordingly (highlighted in cyan).
>
> ---
>
> ### [W1] Relevance of simple datasets
> Thank you for highlighting the concern regarding the relevance of simpler datasets, such as MNIST, in evaluating RelChaNet. While we agree that more complex datasets complement our experiment well (see our response to Q1), we believe that the chosen datasets remain highly relevant due to their alignment with previous studies [1-3]. Experiments in the literature on these datasets demonstrate an advantage of approaches with non-linear capabilities over simpler, more linear methods such as the Fisher Score. From this perspective, we consider them an important benchmark for testing new methods.
>
> We take your concern seriously about the potential for simpler methods to outperform our approach on these datasets. However, we were unable to reproduce the results you mentioned. Specifically, when applying PCA to MNIST with 25 selected features using the approach in [4], we observed a high accuracy of 87.6%, but not the reported 98.0%. Our findings also align with the results reported in [2] for another PCA-based method. If the higher accuracy stems from PCA being used for feature extraction rather than selection, this would explain the discrepancy. Feature extraction, in contrast to feature selection, while also reducing downstream model size, still leverages all features.
>
> Still, the PCA results confirm the existence of simple relationships in the MNIST data and we are thankful for your pointer to include more complex datasets.
>
> [1] Yamada, Yutaro, Ofir Lindenbaum, Sahand Negahban, and Yuval Kluger. “Feature Selection Using Stochastic Gates.” In Proceedings of the 37th International Conference on Machine Learning, edited by Hal Daumé III and Aarti Singh, 119:10648–59. Proceedings of Machine Learning Research. PMLR, 2020. https://proceedings.mlr.press/v119/yamada20a.html.
>
> [2] Lemhadri, Ismael, Feng Ruan, Louis Abraham, and Robert Tibshirani. “LassoNet: A Neural Network with Feature Sparsity.” Journal of Machine Learning Research 22, no. 127 (2021): 1–29.
>
> [3] Liu, Kaiting, Zahra Atashgahi, Ghada Sokar, Mykola Pechenizkiy, and Decebal Constantin Mocanu. “Supervised Feature Selection via Ensemble Gradient Information from Sparse Neural Networks.” In International Conference on Artificial Intelligence and Statistics, 3952–60. PMLR, 2024.
>
> [4] Song, Fengxi, Zhongwei Guo, and Dayong Mei. “Feature Selection Using Principal Component Analysis.” In 2010 International Conference on System Science, Engineering Design and Manufacturing Informatization, 27–30. Yichang, China: IEEE, 2010. https://doi.org/10.1109/ICSEM.2010.14.
>
> ---
>
> ### [W2] Focus on complex datasets
>
> Thanks for the suggestion to focus on datasets with complex, non-linear relationships. We find it a great idea to add more complex datasets to the experiment (see our response to Q1), however, we aim to maintain a wide range of datasets to ensure significant overlap with those previously used in the literature (see our response to W1).
>
> ---
>
> ### [Q1] Performance on complex datasets
> Thank you for suggesting the evaluation of RelChaNet on more complex datasets. We added additional evaluations on the CIFAR-10 and CIFAR-100 datasets in Appendix C. Respecting computational constraints, we limited the analysis to the most relevant alternatives, LassoNet and NeuroFS, and conducted only one run per condition for NeuroFS to keep the analysis under 12 hours per condition. Despite this limitation, we believe these results still provide valuable insights.
>
> The results show that RelChaNet demonstrates competitive performance on these datasets, outperforming both alternatives on CIFAR-10. On CIFAR-100, while RelChaNet significantly outperforms LassoNet, NeuroFS exhibits slightly superior performance. We believe these findings underscore the robustness of RelChaNet for more complex datasets.
>
> ---
>
> Thank you for your help in improving our work. We hope that these updates address your concerns and are looking forward to your feedback.

---

### Author Response · Authors · 2024-11-25
**Global Response**

We thank the reviewers for their constructive comments.

We are encouraged by the reviewers’ recognition of RelChaNet’s novelty. They remarked that it addresses a critical need to reduce computational demand (vxF404) and that its employment of random regrowth likely helps mitigate bias in feature ranking (CZGE04), while being “very simple and easy to implement” (JgCy02).

We are pleased that reviewers found our experiment extensive, spanning across “diverse datasets” (vxF404, HsnL03) as well as a “broad range of competing feature selection methods” (vxF404). RelChaNet shows “strong performance and robustness by outperforming other evaluated methods on 7 out of 9 datasets” (vxF404), which represents a “performance improvement on top of the state-of-the-art” (CZGE04).

Taking into account the reviewers' feedback, we have made the following key changes to improve the paper:

- Added an auxiliary analysis to bolster our evaluation with two more complex datasets (CIFAR-10 and CIFAR-100) and two additional wide datasets (Appendix C)
- Reworked Section 3 with further mathematical treatment of the algorithm, extended rationale on random regrowth, and restructuring to improve readability

We hope these updates address the reviewers' concerns and remain open to further feedback.

---

### Meta-Review · Area_Chair_hQsR · 2024-12-24

**Metareview:**

This paper proposes a supervised feature selection algorithm RelChaNet (and a variant) which uses neuron pruning and regrowth in the input layer to guide the process of feature selection. A normalized score for each input neuron is calculated by aggregating gradients over mini-batches which is then used to update score vector for feature selection.

The proposed approach has been appreciated by most of the reviewers and they also mentioned that the random neuron regrowth might help reducing bias in final ranking of input features as well. However, as rightly pointed out by the reviewers, this work requires improvement in a few crucial aspects as mentioned below:

- careful proofreading and a major rewrite for better readability. A more structured and intuitive presentation would be beneficial. Some theoretical explanations behind the benefits of random neuron regrowth process would make the work solid.
- most reviewers also found the experiments to be rather simplistic. To ensure that the method is effective in feature selection, experiments on more complex tasks would be required. Reviewer CZGE also raised valid concern regarding the poor performance of the method on GLA dataset (contains nearly 50K features) where the proposed method seems to be struggling to meet sota performance and, as already pointed out by the authors, might require carefully designing strategies for improved stopping criterion. Similarly, other valid concerns regarding the suitability of the experiments (monosemanticity etc.) have been raised by the reviewers.

Given the overall feedback, I have to unfortunately reject the work however I'd strongly suggest the authors to incorporate the thoughtful comments provided by the reviewers for a solid submission in future venues.

**Additional Comments On Reviewer Discussion:**

- While the reviewers appreciated the idea behind the work, the major concerns during the rebuttal revolved around (1) the paper writing; (2) lack of experiments on complex tasks (high dim features); and (3) suitability of a few experiments due to the prevalance of monosemanticity in the dataset.
- We appreciate the engagement authors showed during the rebuttal for example (1) improved the readability and phrasing (specifically section 3); (2) provided new experiments on CIFAR10, CIFAR100, BASEHOCH and SMK datasets; (3) discussed the justification behind the random growth etc.
- However, this paper still requires a proper proofreading and better experimental results with ablation and justifications (suggested during the rebuttal) in order to present its utility clearly.

---

### Decision · Program_Chairs · 2025-01-22

Reject